# Pharmacological Effects of Guava (*Psidium guajava* L.) Seed Polysaccharides: GSF3 Inhibits PC-3 Prostate Cancer Cell Growth through Immunotherapy In Vitro

**DOI:** 10.3390/ijms22073631

**Published:** 2021-03-31

**Authors:** Hsiao-Chien Lin, Jin-Yuarn Lin

**Affiliations:** Department of Food Science and Biotechnology, National Chung Hsing University, 250 Kuokuang Road, Taichung 40227, Taiwan; sc11180131@yahoo.com.tw

**Keywords:** *Fas* gene, guava seed, GSF3, pro-(*Bax*)/anti-apoptotic (*Bcl-2*) genes, *Psidium guajava* L., prostate cancer

## Abstract

The inhibitory effects of purified fractions isolated from guava seed polysaccharides (GSPS) including guava seed polysaccharide fraction 1 (GSF1), GSF2, and GSF3 on prostate cancer cells remain unclear. To clarify the anti-prostate cancer potential, GSPS, GSF1, GSF2, and GSF3 were isolated using Sepharose 6B gel filtration chromatography to assay their inhibitory effects on prostate PC-3 cell growth with direct action or indirect immunotherapy using either splenocyte conditioned media (SCM) or macrophage conditioned media (MCM). Correlations between cytokine profiles in the conditioned media and pro-apoptotic gene expression levels in the corresponding treated PC-3 cells were analyzed. Results showed that GSPS, GSF1, GSF2, and GSF3, particularly GSF3, through either direct action or indirect treatments using SCM or MCM, significantly (*p* < 0.05) inhibited PC-3 cell growth. GSF3 direct treatments increased pro-apoptotic Bax/anti-apoptotic Bcl-2 mRNA expression ratios in corresponding treated PC-3 cells. Either SCM or MCM cultured with GSF3 increased Fas mRNA expression levels in corresponding treated PC-3 cells. Both Th2-polarized and anti-inflammatory cytokine IL-10 either secreted in SCM or MCM were positively correlated with Fas mRNA expression levels in corresponding treated PC-3 cells. Our results suggest that GSF3 is a potent biological response modifier to decrease PC-3 cell growth through inducing apoptosis.

## 1. Introduction

Although the cause and risk factors of prostate cancer remain unclear to date and it may not make further progress into the higher mortal stages, prostate cancer is the most commonly suffering cancer in the male population [1]. Inflammation of the prostate gland, e.g., prostatitis, may increase the risk of prostate cancer, and inflammation in the prostate cancer tissue is often found. Even though the link between inflammation and prostate cancer is not easily concluded, anti-inflammation therapy for prostate cancer may be an active research area. Studies indicate that particular cancers may be improved through modulating immune balance to reduce chronic inflammatory reactions with potent materials, such as polysaccharides and other active components, by either direct action or alternative cancer immunotherapy [2,3,4]. Recently, cancer immunotherapy with diverse biological response modifiers (BRMs) including polysaccharides has been developed as a promising anti-cancer method for it could obviously improve adverse effects as a result of chemotherapy and radiotherapy [5]. As potent BRMs, polysaccharides that enhance immunity, but have a low cytotoxicity to host cells, attract much attention for their important role in cancer immunotherapy [3,6,7,8,9,10]. Besides, polysaccharides isolated from fungus, algae, and seeds have been found to have strong pharmacological activities for cancer immunotherapy [11,12,13,14,15]. Most recently, polysulfated polysaccharides heparin like compounds, glycosaminoglycans, in combination with vitamin B17 have been suggested as a possible treatment for prostate cancer [16].

Inducing apoptosis to tumor cells through either intrinsic or extrinsic pathways is an effective tactic for cancer therapy. Intrinsic apoptosis initiates at mitochondria through changing the apoptotic Bcl-2-associated X protein (Bax)/anti-apoptotic B cell CLL/lymphoma 2 (Bcl-2) protein expression ratio [17]. On the contrary, the extrinsic apoptotic pathway starts from membrane death receptor Fas that induces an apoptotic signal by binding to Fas ligand (FasL), which is expressed on the surface of CD8^+^ cytotoxic T cells and subsequently activates caspase cascade in the target cells [18]. *Ganoderma lucidum* Karst. polysaccharides (GLP) have been found to induce Fas protein expression and caspase-3 activation in HCT-116 and LoVo cells, and then sequentially induce apoptosis [19,20]. A polysaccharide isolated from *Gracilariopsis lemaneiformis* Bory de Saint-Vincent was also found to induce apoptosis in A549, MKN28, and B16 cancer cells through increasing Fas/FasL expression in the death receptor pathway [21]. Polysaccharides from different sources may have potential to inhibit differential tumor cells through inducing apoptosis.

Polysaccharides isolated from different sources have been introduced to treat prostate cancers in vitro and *in vivo*. Polysaccharides from *Lycium barbarum* L. inhibit prostate cancer growth through inducing apoptosis in a xenograft mouse model [22]. Nanoyam polysaccharide inhibits prostatic cancer growth through inducing caspase-3 overexpression in PC-3 tumor-bearing mice [23]. A homogeneous polysaccharide with a low molecular weight of 7.0 *×* 10^4^ Da isolated from green tea (*Camellia sinensis* (L.) Kuntze) was found to inhibit PC-3 cell growth via inducing intrinsic apoptosis by elevating Bax/Bcl-2 ratio and caspase-3 protein expression, but decreasing miR-93 [17]. A main pollen polysaccharide CF1 with 1540 kDa molecular weight from Chinese wolfberry inhibits the growth of DU145 prostate cancer cells through the apoptosis process in vitro [24]. Water soluble polysaccharides isolated from *G. lucidum* Karst. inhibit prostate cancer cell migration through a protein arginine methyltransferase 6 (PRMT6) signaling pathway [25,26]. Combined mushroom polysaccharide-K (PSK, extract of *Trametes versicolor* (L.) Lloyd) and docetaxel treatments against human prostate cancer raise tumor-infiltrating CD4^+^ and CD8^+^ T cell numbers and increase the cytolytic activity against YAC-1 cells by splenic natural killer cells in an immunocompetent murine model, suggesting that PSK might decrease tumor growth via apoptosis and immunotherapy [27]. Most recently, novel guava (*Psidium guajava* L.) seed polysaccharides have been characterized for anti-inflammatory and anti-PC-3 prostate cancer potential [2,28]. To date, the use of polysaccharides alone or in combination with other components may be a possible therapy for treating prostate cancer.

Recently, guava seed polysaccharides (GSPS) have been characterized and their inhibitory effects against the growth of prostate and breast cancer cells have been partially disclosed [2,3,5,18,28]. GSPS are composed of three fractions, guava seed polysaccharide fraction 1 (GSF1), GSF2, and GSF3; nevertheless, their anti-prostate cancer potential has not yet been clarified. In the present study, GSPS, GSF1, GSF2, and GSF3 were isolated to analyze their anti-PC-3 potential with direct action and indirect immunotherapy using either splenocyte conditioned media (SCM) or macrophage conditioned media (MCM). Mitochondrial pro-(Bax)/anti-apoptotic (Bcl-2) and membrane death receptor Fas genes expression in the treated PC-3 cells were measured. Correlations between cytokine profiles in the conditioned media and pro-apoptotic gene expression levels in the corresponding treated PC-3 cells were analyzed.

## 2. Materials and Methods

### 2.1. Isolation and Purification of Guava Seed Polysaccharides (GSPS)

A voucher specimen of guava seeds (*Psidium guajava* L.) (www.theplantlist.org accessed on 25 February 2021) used in this study has been deposited in the museum of the Taiwan Agricultural Research Institute (voucher specimen number: 00411585), Kaohsiung, Taiwan. Guava seed polysaccharides (GSPS) were extracted, characterized, and purified using Sepharose 6B gel filtration chromatography as described previously [2,5,18,28]. GSPS further resolved into three fractions, namely guava seed polysaccharide fraction-1 (GSF1), guava seed polysaccharide fraction-2 (GSF2), and guava seed polysaccharide fraction-3 (GSF3), corresponding their molecular weights at 6.1 × 10^8^, 3.3 × 10^7^, and 6.8 × 10^3^ Da, respectively [2,5]. Among GSPS isolated purified fractions, GSF3 is an active proteopolysaccharide for anti-breast cancer cells, consisting of 3.28% of glucuronic acid, 28.13% of galacturonic acid, 14.88% of galactose, 3.96% of mannose, 22.99% of glucose, 7.31% of arabinose, 1.55% of ribose, 14.81% of xylose, 1.68% of fucose, and 1.43% of rhamnose [18,28].

To clarify the anti-prostate cancer effect, isolated GSPS, GSF1, GSF2, and GSF3 were respectively subjected to treat PC-3 prostate cancer cells via either direct action or indirect tumor immunotherapy using immune cell conditioned media [18]. Endotoxin (lipopolysaccharide) contents in the isolated polysaccharides were measured using limulus amoebocyte lysate (LAL) Pyrotell gel clot test (Associations of Cape Cod Incorporated, ACC Catalog #GS003, East Falmouth, MA, USA) before experiments in order to exclude endotoxin contamination. Endotoxin contents in isolated polysaccharides used in this study were <0.03 EU/mg [28].

### 2.2. Preparation of Primary Immune Cell Conditioned Media Using GSPS and Its Purified Fractions GSF1, GSF2, and GSF3

Splenocytes and peritoneal macrophages were isolated from female BALB/c mice and respectively adjusted to 1 × 10^7^ cells/mL and 2 × 10^6^ cells/mL in tissue culture medium (TCM medium) [2,29]. Conditioned media using primary immune cells including splenocytes and macrophages in the absence or presence of GSPS, GSF1, GSF2, and GSF3 at the indicated final optimal concentrations of 0, 8, 40, and 200 μg/mL were prepared and performed as described previously [2,5,18,28]. Cytokine secretion levels in the immune cell conditioned media were determined by enzyme-linked immunosorbent assay (ELISA) [18]. The use protocol of experimental mice was verified and proven (IACUC No: 101-95R) by the Institutional Animal Care and Use Committee, National Chung Hsing University, Taiwan.

### 2.3. Effects of GSPS and Its Purified Fractions GSF1, GSF2, and GSF3 on the Growth of Human Prostate PC-3 Cancer Cells

#### 2.3.1. Culture of PC-3 Cell Line 

The PC-3 cell line was purchased from the Bioresource Collection and Research Center (BCRC, Food Industry Research and Development Institute (FIRDI), Hsinchu, Taiwan, ROC), and cultured as described previously [2]. When PC-3 cells had grown to 90% confluence, cells were harvested with a cell scraper and adjusted to 4 × 10^5^ cells/mL F-12 K medium using a hemocytometer with the trypan blue dye exclusion method for subsequent bio-assay experiments.

#### 2.3.2. Direct Administration Effects of GSPS and Its Purified Fractions GSF1, GSF2, and GSF3 on the Growth of PC-3 Cells

GSPS and its purified fractions GSF1, GSF2, and GSF3 at the indicated concentrations of 0, 16, 80, and 400 μg/mL (50 μL/well) were subjected to treat PC-3 cells (4 × 10^5^ cells/mL, 50 μL/well) in a 96-well plate and incubated at 37 °C in a humidified incubator with 5% CO_2_ and 95% air for 24 and 48 h, respectively. Paclitaxel at the indicated final concentration of 2.5 μΜ was selected as a positive control [2,5,18,28]. The PC-3 cell viability was determined with a 3-(4,5-dimethylthiazol-2-diphenyl)-2,5-tetrazolium bromide (MTT, Sigma, St. Louis, MO, USA) assay. The MTT assay was performed as described previously [2,5,18,28].

#### 2.3.3. Indirect Administration Effects of Primary Immune Cell Conditioned Media Using GSPS and Its Purified Fractions GSF1, GSF2, and GSF3 on the Growth of PC-3 Cells

PC-3 cells (4 × 10^5^ cells/mL, 50 μL/well) were respectively treated with the immune cell conditioned media using GSPS and its purified fractions GSF1, GSF2, and GSF3 (50 μL/well) in a 96-well plate and incubated at 37 °C in a humidified incubator with 5% CO_2_ and 95% air for 24 and 48 h, respectively. Paclitaxel at the indicated final concentration of 2.5 μΜ was selected as a positive control [2,5,18,28]. The PC-3 cell viability (% of control) was determined with an MTT assay [2,5,18,28].

### 2.4. GSF3 Effects on Pro-Apoptotic (Bax), Anti-Apoptotic (Bcl-2), and Fas Gene Expression Levels in PC-3 Cells under Either Direct Action or Indirect Immunotherapy Models

Among GSPS purified fractions, GSF3 showed the most potential to inhibit PC-3 cell growth. Thus, GSF3 was selected to unravel its inhibitory mechanism to PC-3 cell growth targeted at apoptosis. Briefly, PC-3 cells (4 × 10^5^ cells/mL, 3 mL/well) were respectively treated with GSF3 alone at the indicated concentrations of 0, 16, 80, and 400 μg/mL (3 mL/well) and the immune cell conditioned media (2-fold concentrated) using GSF3 (3 mL/well) in 6-well plates. Paclitaxel at a final concentration of 2.5 μΜ was selected as a treatment control in each experiment. The plate was incubated in a humidified incubator with 5% CO_2_ and 95% air at 37 °C for 6 h. The plate was centrifuged at 400× *g* for 10 min to remove the cell culture supernatant. The cell pellet was carefully washed with 1 mL sterile phosphate-buffered saline (137 mM NaCl, 2.7 mM KCl, 8.1 mM Na_2_HPO_4_, 1.5 mM KH_2_PO_4_, pH 7.4, 0.22 μm filtered). Then, the cell pellet in each well was respectively harvested with a cell scraper and collected in a 1.5 mL sterile microcentrifuge tube. The tube was centrifuged at 400*× g* for 10 min to discard the supernatant. Each collected cell pellet was stored at −80 °C for following total RNA extraction. Relative mRNA expression levels of Bax, Bcl-2, and Fas in the treated PC-3 cells were measured using a two-step reverse transcription (RT) and real-time quantitative PCR (qPCR) assay [18,30]. The experiments were performed and expressed as described previously [18,30].

### 2.5. Statistical Analysis

Data are presented as the mean ± standard deviation (SD) and analyzed with one-way analysis of variance (ANOVA), followed by Duncan’s multiple range test by the SPSS system with 20.0. *p* < 0.05 was considered a significant difference among treatments.

## 3. Results

### 3.1. Direct Administration Effects of GSPS and Its Purified Fractions GSF1, GSF2, and GSF3 on PC-3 Cell Growth

GSPS and its purified fractions GSF1, GSF2, and GSF3 were isolated to treat PC-3 prostate cancer cells via both direct action and indirect tumor immunotherapy using immune cell conditioned media. Paclitaxel at the indicated final concentration of 2.5 μM was selected as a positive control. The results showed that paclitaxel significantly (*p* < 0.05) inhibited PC-3 cell viabilities through either 24 or 48 h incubation (Figure 1 and Figure 2). Importantly, GSF1, GSF2, and GSF3 treatments significantly (*p* < 0.05) inhibited PC-3 cell viabilities through 24 and 48 h incubation, particularly 48 h incubation (Figure 1 and Figure 2); however, GSPS administrations just slightly inhibited (*p* > 0.05) PC-3 cell growth at the same time (Figure 1A and Figure 2A). Half maximal (50%) inhibitory concentrations (IC_50_) to PC-3 cells through 48 h incubation for GSF1, GSF2, and GSF3 were estimated at 700, 550, and 320 μg/mL, respectively (Figure 2), suggesting that GSF1, GSF2, and GSF3, particularly GSF3, have direct action to inhibit PC-3 cell growth. GSF3 exerted the most potential to hinder PC-3 cell growth among GSPS purified fractions.

### 3.2. Indirect Administration Effects of GSPS and Its Purified Fractions GSF1, GSF2, and GSF3 on PC-3 Cell Growth Using Immune Cell Conditioned Media

There were no significant differences (*p* > 0.05) in PC-3 cell viabilities between cultured with F-12 K and TCM media (100.0 ± 0.1% versus 94.2 ± 2.8% at 24 h; 102.0 ± 4.8% versus 107.2 ± 10.7% at 48 h), suggesting that TCM medium that existed in immune cell conditioned media would not affect the PC-3 cell growth (Figure 3 and Figure 4). Paclitaxel (a positive control) at 2.5 μM significantly inhibited the PC-3 cell viability incubated for both 24 and 48 h (55.0 ± 12.9% and 52.6 ± 13.3%), suggesting that paclitaxel direct treatment has potential to inhibit PC-3 cell growth (Figure 3). Importantly, treatments with SCM in the absence or presence of polysaccharide fractions through 24 and 48 h incubation significantly (*p* < 0.05) inhibited the PC-3 cell viability (12.2 ± 2.8–29.4 ± 6.5%) as compared to those of controls (cells alone, about 100%), suggesting that splenocyte secretions (such as Th1/Th2 cytokines) in the tumor microenvironment may inhibit PC-3 cell growth (Figure 3). Most importantly, SCM cultured in the presence of GSPS and GSF3, particularly GSF3, significantly (*p* < 0.05) enhanced their inhibitory effects on PC-3 cell viabilities as compared to those of controls, respectively (Figure 3). However, SCM cultured with GSF1 and GSF2 at different concentrations just slightly (*p* > 0.05) enhanced their inhibitory effects on PC-3 cell growth as compared to those of controls. In comparison with differential effects of GSPS purified fractions, our results suggest that GSF3 is a major active component in GSPS to inhibit PC-3 cell growth through enhancing the inhibitory effect of SCM, possibly through modulating Th1/Th2 cytokine secretion profiles of splenocytes.

Similar to the effects of SCM, treatments with MCM in the absence or presence of polysaccharide fractions through 24 and 48 h incubation significantly (*p* < 0.05) inhibited PC-3 cell viabilities as compared to those of controls (cells alone in F-12K or TCM medium) (Figure 4). Our results suggest that macrophage secretions (such as M1/M2 cytokines) in the tumor microenvironment may inhibit PC-3 cell growth. Most importantly, MCM in the presence of GSPS and GSF3, particularly GSF3, significantly (*p* < 0.05) enhance their inhibitory effects on PC-3 cell viabilities as compared to those of controls through 24 and 48 h incubation, respectively (Figure 4). However, treatments with MCM in the presence of GSF1 and GSF2 at different concentrations could not significantly (*p* > 0.05) enhance their inhibitory effects on PC-3 cell viabilities as compared to those of controls. Our results evidenced that GSF3 had the most potential to inhibit PC-3 cell growth through enhancing the effect of MCM, possibly through modulating M1/M2 cytokine secretion of macrophages.

In comparison with the SCM and MCM effects on PC-3 cell growth, we concluded that both SCM and MCM have profound effects against PC-3 cell growth (Figure 3 and Figure 4). The results further suggest that secretions (such as cytokines) by immune cells including splenocytes (most are T and B cells) and macrophages, which may exist in the tumor microenvironment, may inhibit prostate cancer cell growth. Most importantly, GSF3 exerted the most inhibitory activity to PC-3 cell growth through direct action and indirect tumor immunotherapy (Figure 1, Figure 2, Figure 3 and Figure 4). Therefore, GSF3 was further selected to treat PC-3 cells with direct action and indirect immunotherapy to clarify a possible mechanism targeted at an intrinsic and extrinsic apoptotic pathway.

### 3.3. GSF3 Treatment Effects on Bax and Bcl-2 mRNA Expression Levels in PC-3 Cells through Either Direct Action or Indirect Immunotherapy Using Immune Cell Conditioned Media

To examine whether GSF3 exerted its pro-apoptotic effects through modulating mitochondrial Bax and Bcl-2, as well as membrane death receptor Fas, GSF3 was selected to treat PC-3 cells through direct action and indirect immunotherapy for 6 h. Paclitaxel was selected a positive control in the experiment. Our data showed that paclitaxel direct treatment significantly (*p* < 0.05) increased pro-apoptotic Bax mRNA expression levels and Bax/Bcl-2 mRNA expression ratio in the PC-3 cells as compared to that in the vehicle control (VC) (Table 1), suggesting that paclitaxel direct treatment may inhibit PC-3 cell growth through promoting cancer cell apoptosis. GSF3 direct treatments significantly (*p* < 0.05) and dose-dependently increased both Bax and Bcl-2 mRNA expression levels in the PC-3 cells (Table 1). Importantly, Bax/Bcl-2 mRNA expression ratios in the corresponding treated PC-3 cells were significantly (*p* < 0.05) increased by GSF3 direct treatments, evidencing that there is a pro-apoptotic effect on PC-3 cells with GSF3 direct treatment.

As to GSF3 indirect treatment effects, SCM alone (without GSF3) significantly (*p* < 0.05) increased the anti-apoptotic Bcl-2 mRNA expression amount, but decreased the Bax/Bcl-2 mRNA expression ratio in the treated PC-3 cells as compared to those in the VC (Table 1). The results implied that the treatment with SCM alone (without GSF3) could not inhibit PC-3 cell apoptosis via a decreasing mitochondrial Bax/Bcl-2 expression ratio. However, SCM cultured with GSF3 (SGSF3) slightly (*p* > 0.05) increased Bax/Bcl-2 mRNA expression ratios in the treated PC-3 cells as compared to that of SCM alone (without GSF3). SGSF3 treatments still could not totally reverse these mitochondrial apoptotic components in the treated PC-3 cells if compared to those in the vehicle control. We supposed that other signaling, except mitochondrial Bax/Bcl-2 ratios, might be involved in the inhibition of PC-3 cell growth by SCM; possibly Fas, which is a membrane death receptor.

Similar to the effect of SCM, macrophage conditioned media (MCM) alone (without GSF3) significantly (*p* < 0.05) decreased Bax/Bcl-2 mRNA expression ratios compared to that in the VC, suggesting that the treatment with MCM alone (without GSF3) could not inhibit PC-3 cell apoptosis via decreasing the mitochondrial Bax/Bcl-2 expression ratio (Table 1). Importantly, treatments with MCM cultured with GSF3 (MGSF3) significantly (*p* < 0.05) increased Bax/Bcl-2 mRNA expression ratios in the treated PC-3 cells compared to that of MCM alone (without GSF3). Even though, MGSF3 treatments still could not surpass these mitochondrial apoptotic components in the treated PC-3 cells if compared to those in the vehicle control. We hypothesized that other signaling, except mitochondrial Bax/Bcl-2 ratios, might be involved in the inhibition of PC-3 cell growth by MCM, possibly membrane death receptor Fas. Therefore, changes in Fas mRNA expression levels in the treated PC-3 cells were measured. 

### 3.4. GSF3 Treatment Effects on Fas mRNA Expression Levels in Treated PC-3 Cells through Either Direct Action or Indirect Immunotherapy Using Immune Cell Conditioned Media

The results showed that paclitaxel direct treatment at 2.5 μM slightly, but not significantly (*p* > 0.05), increased Fas mRNA expression levels in the treated PC-3 cells as compared to that of the vehicle control (Table 2). Importantly, GSF3 direct treatments significantly (*p* < 0.05) and dose-dependently increased Fas mRNA expression levels in the treated PC-3 cells, suggesting that GSF3 direct treatments enhance the opportunity of PC-3 cells’ apoptosis via the extrinsic pathway through the membrane death receptor Fas. Splenocyte (SCM) and macrophage conditioned media (MCM) alone (without GSF3) could not significantly (*p* > 0.05) influence Fas mRNA expression levels in the treated PC-3 cells compared to the vehicle control. Most importantly, treatments with SGSF3 and MGSF3, which were cultured with 200 μg/mL GSF3, significantly (*p* < 0.05) increased Fas mRNA expression levels in the treated PC-3 cells compared to those of SCM and MCM alone (without GSF3), respectively. Increased Fas mRNA expression levels in the treated PC-3 imply that PC-3 cells may have the chance to receive more death signals to induce apoptosis. Our results suggest that GSF3 indirect immunotherapies using SCM and MCM may enhance the pro-apoptotic effect, possibly via changing the cytokine secretion profile in the immune cell conditioned media. In the present study, GSF3, SGSF3, and MGSF3 have been evidenced to effectively increase Fas mRNA expression levels in the treated cells that may consequently enhance PC-3 cells’ apoptosis. We supposed that the cytokine secretion profile by immune cells may be correlated with Fas mRNA expression levels in the corresponding treated PC-3 cells. Therefore, correlations between cytokine secretion profiles in immune cell conditioned media cultured without or with GSF3 and Fas mRNA expression levels in the corresponding treated PC-3 cells were further analyzed.

### 3.5. Correlations between Cytokine Secretion Profiles in Immune Cell Conditioned Media Cultured without or with GSF3 and Fas mRNA Expression Levels in the Corresponding Treated PC-3 Cells 

To correlate the cytokine secretion profile by immune cells (splenocytes and macrophages) and Fas mRNA expression levels in the corresponding treated PC-3 cells, the relationships were respectively analyzed using Pearson’s correlation coefficients (r). We observed that there was no significant correlation (r = 0.140, *p* = 0.566) between IL-2 (Th1 cytokine) secretion levels in SGSF3 media and Fas mRNA expressions in the corresponding treated PC-3 cells (Figure 5A). Most importantly, there are significant (*p* < 0.05) positive correlations between IL-10 (Th2 cytokine) secretion levels (r = 0.672, *p* = 0.001; Figure 5B), as well as IL-10/IL-2 (Th2/Th1 cytokine secretion ratio) (r = 0.671, *p* = 0.001; Figure 5C) and Fas mRNA expression levels in the corresponding treated PC-3 cells. It is suggested that Th2-polarized cytokines secreted by splenic T cells in the PC-3 tumor microenvironment may increase Fas mRNA expression levels in the corresponding treated PC-3 cells, possibly subsequently causing PC-3 cells’ apoptosis induced by Fas ligand (FasL)-bearing cells such as CD8^+^ T cells.

Furthermore, IL-1β, IL-6, TNF-α, IL-10, and (IL-6+TNF-α)/IL-10 cytokine secretion profiles in MGSF3 media were correlated with Fas mRNA expression levels in the corresponding treated PC-3 cells using Pearson’s correlation coefficients (r) (Figure 6). There are significant (*p* < 0.05) positive correlations between IL-1β (r = 0.447, *p* = 0.048; Figure 6A), IL-6 (r = 0.482, *p* = 0.032; Figure 6B), TNF-α (r = 0.498, *p* = 0.025; Figure 6C), as well as IL-10 (r = 0.748, *p* = 0.000; Figure 6D) secretion amounts in the media of MGSF3 and Fas mRNA expression levels in the corresponding treated PC-3 cells. Consequently, IL-10 (an anti-inflammatory and macrophage type 2 (M2) cytokine) among cytokines secreted by macrophages exhibited a dominant effect with the highest r value but the lowest *p* value (Figure 6D). Moreover, there is a negative correlation between (IL-6+TNF-α)/IL-10 (r = −0.394, *p* = 0.086; Figure 6E) (pro-/anti-inflammatory cytokines) secretion ratios and Fas mRNA expression levels in the media of MGSF3 and Fas mRNA expression levels in the corresponding treated PC-3 cells. Our results evidenced that increased anti-inflammatory cytokines secreted by macrophages might increase Fas mRNA expressions in the treated cancer cells. The present study concluded that Th2-polarized and anti-inflammatory cytokine IL-10 either secreted by T lymphocytes in the splenocytes (Figure 5) or peritoneal macrophages (Figure 6) may induce Fas mRNA expression levels in the treated PC-3 cells. Anti-inflammatory IL-10 may be selected as an anti-cancer agent for treating prostate cancer.

## 4. Discussion

Polysaccharides from fungus, algae, and other plants are reported to have strong pharmacological activities through modulating immune balance and may treat cancers via immunotherapy [11,12,13,14,15]. To investigate pharmacological effects of guava seed polysaccharides and achieve optimal concentrations for bioassays, GSPS, GSF1, GSF2, and GSF3 at 0, 1.6, 8, 40, 200, and 500 μg/mL were respectively administered to primary splenocytes for 72 hrs. Treatments with GSPS, GSF1, GSF2, and GSF3 at 1.6–500 μg/mL did not significantly cause any cytotoxicity to splenocytes. To compare concisely and effectively, GSPS, GSF1, GSF2, and GSF3 at 8, 40, and 200 μg/mL were recommended as optimal concentrations to primary immune cells and used for bioassay experiments [28]. We have evidenced that the adopted concentrations of 8, 40, and 200 μg/mL are sufficient to compare the bioactivities of these polysaccharide fractions [18]. In the present study, GSPS, GSF1, GSF2, and GSF3 direct or indirect treatments, particularly GSF3, significantly inhibited PC-3 cell viability incubated for either 24 or 48 h incubation (Figure 1, Figure 2, Figure 3 and Figure 4), evidencing that GSF3 has the most potential to inhibit prostate cancer. We supposed that GSF3 may play a vital role as BRMs, which are potent compounds for treating PC-3 cells by changing naturally occurring immune processes [6,7,8,10]. GSF3, which is a novel polysaccharide isolated from guava (*P. guajava* L.) seeds, has been characterized and found to inhibit MCF-7 breast cancer cell growth through increasing either the Bax/Bcl-2 ratio or Fas mRNA levels in the target cancer cells [18,28]. Papaya (*Cydonia oblonga* Miller) polysaccharides increased *Bax*, but inhibited *Bcl-2* gene expression in HCT-116 cells, leading to apoptosis due to cytochrome c release and increased caspase-3 activity [31]. A polysaccharide isolated from *Angelica sinensis* (Oliv.) Diels (ASP) induces breast cancer cells apoptosis by influencing PARP, Bax, Bcl-2, Bcl-xL, and Apaf-1 protein expression in human breast T47D and Hs578T cells through caspase-3 signaling pathways [32]. In the present study, we further evidenced that direct treatments with GSF3 enhanced pro-apoptotic effects via increasing Bax/Bcl-2 mRNA expression ratios in the treated PC-3 cells (Table 1). However, our results also exhibited that other apoptotic signaling pathways, in addition to the mitochondrial Bax/Bcl-2 expression ratio, might be involved in the inhibition to PC-3 cell growth when tumor immunotherapy, such as SCM or MCM, was adopted (Table 1). We hypothesized that Fas, an apoptosis-inducing death receptor on the cell membrane, might partake in the process. Importantly, we evidenced that GSF3, SGSF3, and MGSF3 might effectively increase PC-3 cells apoptosis via increasing Fas mRNA expression levels in the treated cells (Table 2). Elevating Fas expression can enhance the recognition chance by FasL and then promote PC-3 cell apoptosis. Consequently, tumor-infiltrating CD8+ T cells or NK cells may be recruited to inhibit PC-3 cell growth via increasing apoptosis and antitumor responses by membrane FasL on particular immune cells [27].

In the present study, Th2-polarized and anti-inflammatory cytokine IL-10 either secreted by T lymphocytes in the splenocytes or peritoneal macrophages in the tumor microenvironment were correlated with Fas mRNA expression levels in the treated PC-3 cells (Figure 5 and Figure 6) [2,18,28,33]. Our results further suggest that IL-10 may be further selected as an anti-prostate cancer agent (Table 2, Figure 5 and Figure 6). Moreover, GSF3 treatments have been proven to induce Th2-polarized cytokine secretion by splenocytes [18]. In comparison to the effect of GSPS against PC-3 cells via tumor immunotherapy, GSF3 obviously increased its inhibitory effect via decreasing pro-inflammatory/anti-inflammatory cytokine secretion ratios in the tumor microenvironment, suggesting that GSF3 is the major active fraction in GSPS to inhibit PC-3 cell growth [2]. Cytokine tumor immunotherapy through modulating cytokine secretion profiles by potent polysaccharide GSF3 may be helpful to health and is a promising method to treat prostate cancer.

Guava (*P. guajava* L.) is traditionally used like food and in folk medicine, including infusions and decoctions of guava’s root, bark, leaves, fruits, and/or seeds for oral and topical use for treating many diseases including cancers [34,35,36]. However, making use of guava seeds for health purposes has attracted much attention in recent years [2,3,18,28,37]. Crude guava seed polysaccharides (GSPS) were found to inhibit the growth of human prostate cancer PC-3 cells through decreasing pro-inflammatory/anti-inflammatory cytokine secretion ratios by macrophages in a tumor microenvironment [2]. After being purified and characterized, GSF3 was found to inhibit MCF-7 breast cancer cell growth via the decreasing of Bcl-2 mRNA expression levels but the increasing of pro-(Bax)/anti-apoptotic (Bcl-2) mRNA expression ratios in the treated cells [18,28]. Previous findings have evidenced that GSF3 is a potential anti-cancerous polysaccharide to inhibit MCF-7 breast cancer cell growth through modulating cytokine secretion profiles of immune cells [18]. However, active purified polysaccharides from guava seeds (GSPS) targeted at prostate cancer treatments remain unclear; therefore, it was further investigated in the present study. GSPS, GSF1, GSF2, and GSF3 have been carefully and sequentially characterized with an immunomodulatory activity [28]. There are three kinds of polysaccharides including simple polysaccharides, glycoproteins, and proteopolysaccharides. We evidenced that all of these guava seed polysaccharides are proteopolysaccharides in our previous study [28]. Characteristics of a proteopolysaccharide are thermostable and sugar-rich, which are properties that are similar to a simple polysaccharide but not a protein, even though a proteopolysaccharide covers a protein moiety. The thermostable and sugar-rich properties of GSPS, GSF1, GSF2, and GSF3 may have practical applications in pharmacological effects, particularly GSF3 for inhibiting PC-3 prostate cancer cell growth through immunotherapy.

Our results have attained some achievements to evidence that GSF3 hindered PC-3 prostate cancer cell growth through enhancing its pro-apoptotic gene expressions, possibly via changing Th2- and anti-inflammatory cytokine secretions by immune cells. However, there are some limitations for GSF3 application. Firstly, it is still an in vitro study; to confirm the key findings, an animal model xerografted with human prostate cancer cells for testing GSF3 function has been under study. More biomarkers of PC-3 cells modulated by GSF3 treatments in vitro and in vivo might be determined to globally confirm apoptotic mechanisms. The expression of galectin-3 and galectin-9 on PC-3 cells that are proteins binding to β-galactoside sugars to provide potential biomarkers and therapeutic targets in particular cancer should be checked in the future [38,39]. Moreover, the GSF3 polysaccharides with low molecular weight should be further characterized for their charges using an anion exchange column because negative charged polysaccharides may have more immunoactivities than those of neutral equivalents.

## 5. Conclusions

In this study, only GSF3 among guava seed polysaccharide components inhibited PC-3 cell growth using both direct treatments and indirect SCM and MCM. GSF3 direct treatments increased pro-(Bax)/anti-apoptotic (Bcl-2) mRNA expression ratios in the corresponding treated PC-3 cells. SCM and MCM cultured with GSF3 increased Fas mRNA expression in the corresponding treated PC-3 cells. Both Th2-polarized and anti-inflammatory cytokine IL-10 either secreted by T lymphocytes in the splenocytes or peritoneal macrophages were correlated with Fas mRNA expression levels in the corresponding treated PC-3 cells. It is concluded that a novel GSF3 inhibited PC-3 cell growth, which possibly resulted from PC-3 apoptosis through increasing *Bax*/*Bcl-2* gene expression ratios and *Fas* gene expression via either direct action or immunotherapy to achieve its anticancer effects. GSF3 would be expected to be a potential source of health foods and pharmaceuticals to treat the prostate cancer via its potent immunomodulatory activity.

## Figures and Tables

**Figure 1 ijms-22-03631-f001:**
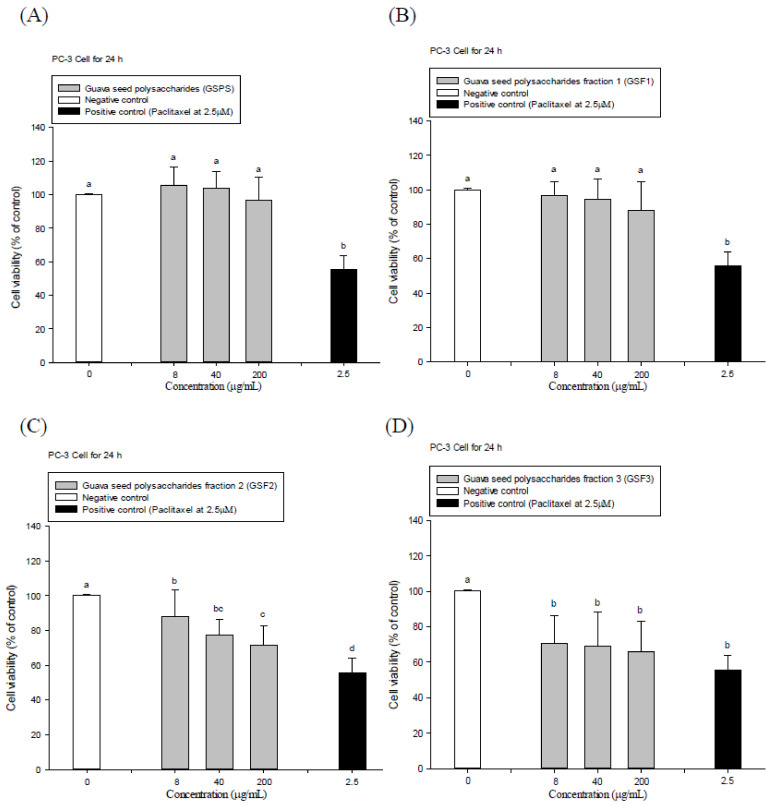
Effects of direct treatments with guava seed polysaccharides (GSPS) (**A**), GSF1 (**B**), GSF2 (**C**), and GSF3 (**D**) for 24 h on PC-3 cell growth. Values are means ± SD (n = 6 biological determinations). Bars in the plot not sharing a common letter are significantly different (*p* < 0.05) from each other analyzed by one-way ANOVA, followed by Duncan’s multiple range test. Each cell population (2 × 10^5^ cells/mL) was respectively treated with the polysaccharides at the indicated concentrations of 8, 40, and 200 μg/mL, as well as paclitaxel (a positive control) at 2.5 μM. The cell viability was determined using MTT assay.

**Figure 2 ijms-22-03631-f002:**
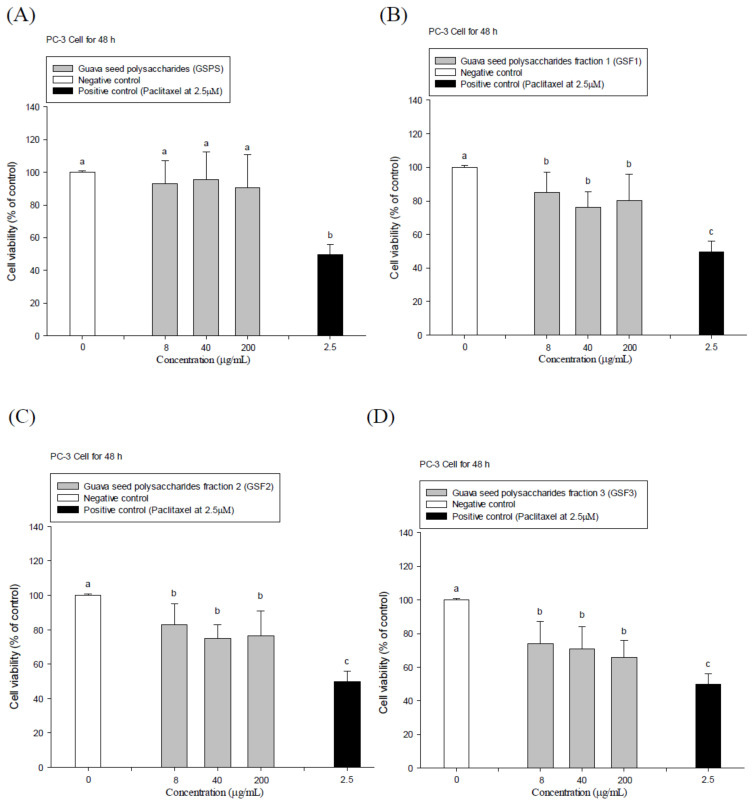
Effects of direct treatments with guava seed polysaccharides (GSPS) (**A**), GSF1 (**B**), GSF2 (**C**), and GSF3 (**D**) for 48 h on PC-3 cell growth. Values are means ± SD (n = 6 biological determinations). Bars in the plot not sharing a common letter are significantly different (*p* < 0.05) from each other analyzed by one-way ANOVA, followed by Duncan’s multiple range test. Each cell population (2 × 10^5^ cells/mL) was respectively treated with the polysaccharide at the indicated concentrations of 8, 40, and 200 μg/mL, as well as paclitaxel (a positive control) at 2.5 μM. The cell viability was determined using MTT assay.

**Figure 3 ijms-22-03631-f003:**
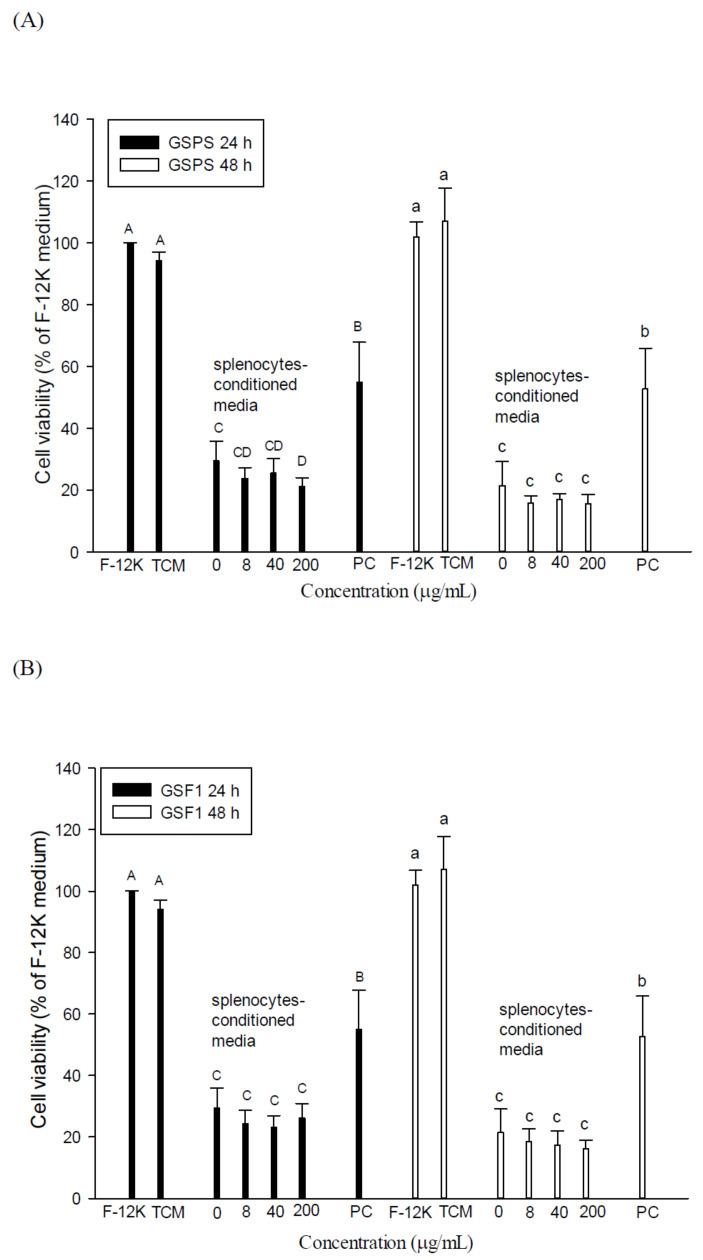
Effects of indirect treatments with splenocyte conditioned media (SCM) cultured with guava seed polysaccharide (GSPS) (**A**), GSF1 (**B**), GSF2 (**C**), and GSF3 (**D**) on PC-3 cell growth. Cells (2 × 10^5^ cells/mL) were treated with SCM for 24 and 48 h, respectively. Values are means ± SD (n = 6 biological determinations). Bars in the same plot at the same incubation time not sharing a common letter are significantly different (*p* < 0.05) from each other, assayed by one-way ANOVA, followed by Duncan’s multiple range test. SCM, conditioned media of splenocytes cultured with different polysaccharide fractions at different concentrations; PC, positive control (paclitaxel at 2.5 μΜ).

**Figure 4 ijms-22-03631-f004:**
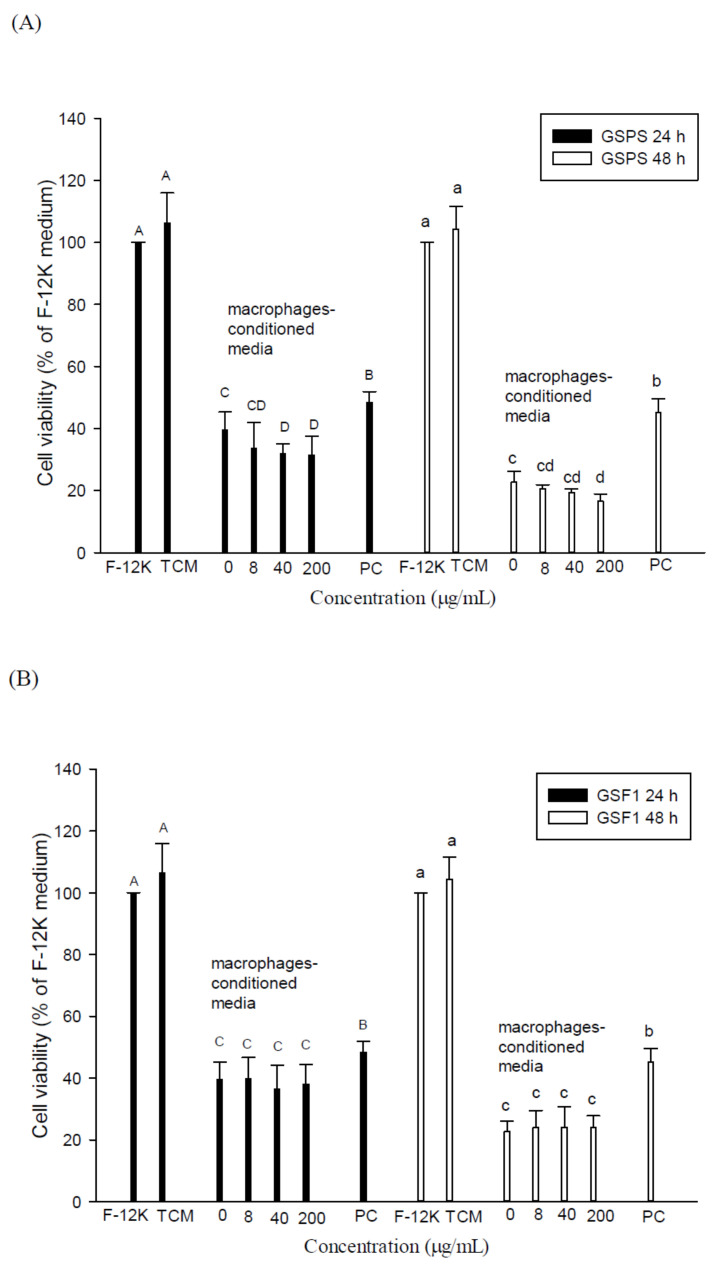
Effects of indirect treatments with macrophage conditioned media (MCM) cultured with guava seed polysaccharide (GSPS) (**A**), GSF1 (**B**), GSF2 (**C**), and GSF3 (**D**) on PC-3 cell growth. Cells (2 × 10^5^ cells/mL) were treated with MCM for 24 and 48 h, respectively. Values are means ± SD (n = 6 biological determinations). Bars in the same plot at the same incubation time not sharing a common letter are significantly different (*p* < 0.05) from each other, assayed by one-way ANOVA, followed by Duncan’s multiple range test. MCM, conditioned media of peritoneal macrophages cultured with polysaccharide fractions at different concentrations; PC, positive control (paclitaxel at 2.5 μΜ).

**Figure 5 ijms-22-03631-f005:**
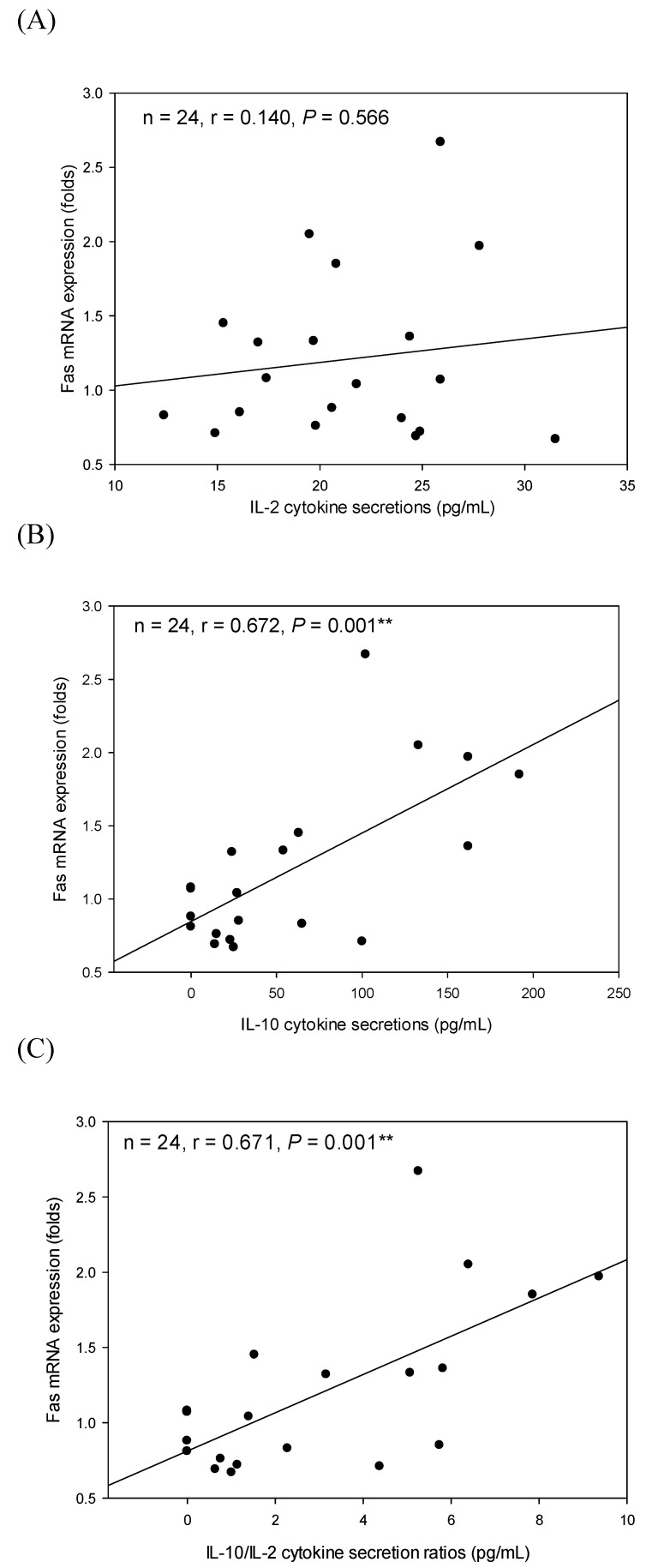
Correlations between IL-2 (**A**), IL-10 (**B**), as well as IL-10/IL-2 (**C**) cytokine secretion profiles in splenocyte conditioned media cultured with GSF3 (SGSF3) and Fas mRNA expression amounts in the corresponding treated PC-3 cells. The correlation was expressed by Pearson product-moment correlation coefficient (r). The correlation is considered statistically different if *p* < 0.05. **, *p* < 0.01.

**Figure 6 ijms-22-03631-f006:**
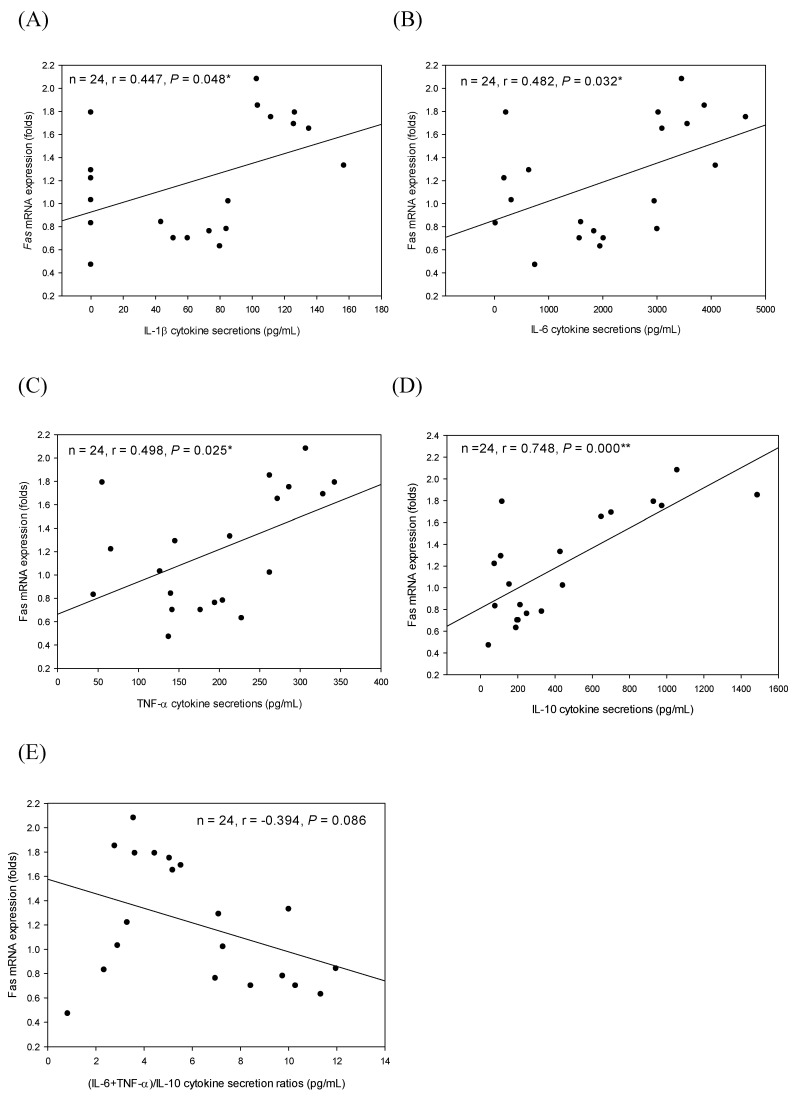
Correlations between IL-1β (**A**), IL-6 (**B**), TNF-α (**C**), IL-10 (**D**), as well as (IL-6+TNF-α)/IL-10 (**E**) cytokine secretion profiles in macrophage conditioned media cultured with GSF3 (MGSF3) and Fas mRNA expression amounts in the corresponding treated PC-3 cells. The correlation was expressed by the Pearson product-moment correlation coefficient (r). The correlation is considered statistically different if *p* < 0.05. *, *p* < 0.05; **, *p* < 0.01.

**Table 1 ijms-22-03631-t001:** Effects of GSF3 direct treatments and indirect immunotherapy using splenocyte conditioned media (SGSF3) and macrophage conditioned media (MGSF3) cultured with GSF3 on Bax and Bcl-2 mRNA expression levels in human prostate cancer PC-3 cells.

Treatments	Conc. (μg/mL).	Relative Expression (Fold)		
		Bax	Bcl-2	Bax/Bcl-2
VC	0	1.02 ± 0.25 ^CDE^	1.00 ± 0.07 ^DE^	1.02 ± 0.24 ^D^
GSF3	8	1.24 ± 0.22 ^BC^	0.85 ± 0.15 ^E^	1.43 ± 0.12 ^B^
	40	1.52 ± 0.05 ^B^	1.15 ± 0.08 ^D^	1.32 ± 0.08 ^BC^
	200	2.36 ± 0.84 ^A^	1.42 ± 0.34 ^C^	1.67 ± 0.20 ^A^
SGSF3	0 8 40 200	0.77 ± 0.16^DE^	1.90 ± 0.35 ^B^	0.40 ± 0.05 ^H^
		1.16 ± 0.26 ^BCD^	2.29 ± 0.12 ^A^	0.51 ± 0.14 ^GH^
		0.82 ± 0.07 ^CDE^	1.06 ± 0.15 ^DE^	0.79 ± 0.13 ^EF^
		0.68 ± 0.14 ^E^	1.12 ± 0.15 ^DE^	0.61 ± 0.21 ^FG^
MGSF3	0 8 40 200	0.90 ± 0.09 ^CDE^	1.29 ± 0.08 ^CD^	0.70 ± 0.08 ^FG^
		1.52 ± 0.10 ^B^	1.44 ± 0.31 ^C^	1.08 ± 0.15 ^D^
		1.18 ± 0.28 ^BCD^	1.03 ± 0.11 ^DE^	1.14 ± 0.14 ^CD^
		1.02 ± 0.22 ^CDE^	1.08 ± 0.14 ^DE^	0.96 ± 0.20 ^DE^
Paclitaxel	2.5 μM	1.49 ± 0.24 ^B^	1.09 ± 0.16 ^DE^	1.37 ± 0.15 ^B^

Cells (2 × 10^5^ cells/mL) were treated with samples for 6 h. Values are mean ± SD (n = 6 biological determinations). Values within the same column not sharing a common superscript capital letter are significantly different (*p* < 0.05) from each other, analyzed by one-way ANOVA, followed by Duncan’s multiple range test. VC, vehicle control (cell line alone); GSF3, guava seed fraction 3; SGSF3, splenocyte conditioned media cultured with GSF3; MGSF3, macrophage conditioned media cultured with GSF3; paclitaxel, a positive control.

**Table 2 ijms-22-03631-t002:** Effects of GSF3 direct treatments and indirect immunotherapy using splenocyte conditioned media (SGSF3) and macrophage conditioned media (MGSF3) cultured with GSF3 on Fas mRNA expression levels in human prostate cancer PC-3 cells.

Treatments	Conc. (μg/mL)	Relative Expression (Fold)
		Fas
VC	0	1.02 ± 0.22 ^C^
GSF3	8	1.09 ± 0.11 ^C^
	40	1.95 ± 0.47 ^B^
	200	2.92 ± 0.49 ^A^
SGSF3	0	1.08 ± 0.32 ^C^
	8	0.96 ± 0.14 ^C^
	40	0.78 ± 0.15 ^C^
	200	1.98 ± 0.47 ^B^
MGSF3	0	1.10 ± 0.45 ^C^
	8	0.73 ± 0.08 ^C^
	40	1.04 ± 0.27 ^C^
	200	1.80 ± 0.16 ^B^
Paclitaxel	2.5 μM	1.07 ± 0.32 ^C^

Cells (2 × 10^5^ cells/mL) were treated with samples for 6 h. Values are mean ± SD (n = 6 biological determinations). Values within the same column not sharing a common superscript capital letter are significantly different (*p* < 0.05) from each other, analyzed by one-way ANOVA, followed by Duncan’s multiple range test. VC, vehicle control (cell line alone); GSF3, guava seed fraction 3; SGSF3, splenocyte conditioned media cultured with GSF3; MGSF3, macrophage conditioned media cultured with GSF3; paclitaxel, a positive control.

## Data Availability

The datasets used and/or analyzed during the current study available from the corresponding author on reasonable request.

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
