# Peer review of "Pharmacological Effects of Guava (Psidium guajava L.) Seed Polysaccharides: GSF3 Inhibits PC-3 Prostate Cancer Cell Growth through Immunotherapy In Vitro"

_ijms, 2021, doi:10.3390/ijms22073631_

Round 1

Reviewer 1 Report

The Review of the manuscript “Pharmacological effects of guava seed polysaccharides: GSF-3 inhibits PC-3 prostate cancer cell growth through immunotherapy in vitro”

The Authors of the manuscript deal with anticancer activity of guava seed polysaccharides fractions (GSF1, GSF2 and GSF3) in treating PC-3 cells. The scientific novelty is good, technical quality is fine but some information should be added.

The guava seed polysaccharides was – GSPS also used in Authors manuscript from 2016 also on the same PC-2 cells (Integrative Cancer Therapies 15(4)). GSF1, GSF2,  GSF3 fractions should be characterised. Polysaccharides could be simple polysaccharides, glycoproteins,and proteopolysaccharides. These compounds should be determined in all fractions of guava seeds.

Author Response

Response to Reviewer 1 Comments

Ms. Ref. No.: ijms-1142027

Point 1: The Authors of the manuscript deal with anticancer activity of guava seed polysaccharides fractions (GSF1, GSF2 and GSF3) in treating PC-3 cells. The scientific novelty is good, technical quality is fine but some information should be added.

Response 1: We thank the Reviewer’s positive comments. Responses to the Reviewer’s comments together with the itemized changes have been made.

Point 2: The guava seed polysaccharides was – GSPS also used in Authors manuscript from 2016 also on the same PC-3 cells (Integrative Cancer Therapies 15(4)). GSF1, GSF2, GSF3 fractions should be characterised. Polysaccharides could be simple polysaccharides, glycoproteins, and proteopolysaccharides. These compounds should be determined in all fractions of guava seeds.

Response 2: Page 28 lines 6 - 15 (revised manuscript): We thank the Reviewer’s remind. In fact, GSPS, GSF1, GSF2 and GSF3 have been carefully and sequentially characterized with an immunomodulatory activity [28]. There are three kinds of polysaccharides including simple polysaccharides, glycoproteins and proteopolysaccharides. We evidenced that all of these guava seed polysaccharides are proteopolysaccharides in our previous study [28]. Characteristics of a proteopolysaccharide are thermostable and sugar-rich which properties are similar to a simple polysaccharide but not a protein, even though a proteopolysaccharide covers a protein moiety. The thermostable and sugar-rich properties of GSPS, GSF1, GSF2 and GSF3 may have practical applications in pharmacological effects, particularly GSF3 for inhibiting PC-3 prostate cancer cell growth through immunotherapy. The description “GSPS, GSF1, GSF2 and GSF3 have been carefully and sequentially characterized with an immunomodulatory activity [28]. There are three kinds of polysaccharides including simple polysaccharides, glycoproteins and proteopolysaccharides. We evidenced that all of these guava seed polysaccharides are proteopolysaccharides in our previous study [28]. Characteristics of a proteopolysaccharide are thermostable and sugar-rich which properties are similar to a simple polysaccharide but not a protein, even though a proteopolysaccharide covers a protein moiety. The thermostable and sugar-rich properties of GSPS, GSF1, GSF2 and GSF3 may have practical applications in pharmacological effects, particularly GSF3 for inhibiting PC-3 prostate cancer cell growth through immunotherapy.” has been added into the revised manuscript.

Reviewer 2 Report

For all bioassay experiments, the concentration of the GSPS fractions, 1,2,3 should be adjusted to reflect their relative concentration in GSPS so that a fair comparison of their activity relative to the activity of GSPS is possible.

I would suggest taking a broad range of concentrations, some in between 40 and 200ug/ml and some even higher.

Author Response

Ms. Ref. No.: ijms-1142027

Comments and Suggestions for Authors

For all bioassay experiments, the concentration of the GSPS fractions, 1,2,3 should be adjusted to reflect their relative concentration in GSPS so that a fair comparison of their activity relative to the activity of GSPS is possible.

I would suggest taking a broad range of concentrations, some in between 40 and 200ug/ml and some even higher.

Response: Page 26 lines 4 - 12 (revised manuscript): We thank the Reviewer’s suggestion. In fact, to achieve optimal concentrations to treat primary immune cells, primary splenocytes from BALB/c mice were administered with isolated GSPS and its purified polysaccharide fractions GSF1, GSF2 and GSF3 at the indicated final concentrations of 0, 1.6, 8, 40, 200 and 500 μg/ml for 72 hours, respectively (as the attached Figure 1). Our results showed that all of GSPS, GSF1, GSF2 and GSF3 administrations at the indicated concentrations did not significantly (P > 0.05) result in any cytotoxicity to splenocytes compared to that of the control, respectively. It was concluded that isolated GSPS, GSF1, GSF2 and GSF3 at the indicated concentrations did not cause any cytotoxicity to splenocytes. To be more concise and effective, GSPS, GSF1, GSF2 and GSF3 at 8, 40 and 200 μg/ml were suggested to be optimal concentrations to primary immune cells and used for the following bioassay experiments [28]. We have evidenced that the adopted concentrations between 8 and 200 μg/ml are sufficient to compare the bioactivities of these polysaccharide fractions [18]. The description “To investigate pharmacological effects of guava seed polysaccharides and achieve optimal concentrations for bioassays, GSPS, GSF1, GSF2 and GSF3 at 0, 1.6, 8, 40, 200 and 500 μg/ml were respectively administered to primary splenocytes for 72 hrs. Treatments with GSPS, GSF1, GSF2 and GSF3 at 1.6 - 500 μg/ml did not significantly cause any cytotoxicity to splenocytes. To compare concisely and effectively, GSPS, GSF1, GSF2 and GSF3 at 8, 40 and 200 μg/ml were recommended as optimal concentrations to primary immune cells and used for bioassay experiments [28]. We have evidenced that the adopted concentrations of 8, 40 and 200 μg/ml are sufficient to compare the bioactivities of these polysaccharide fractions [18].” has been added in the revised manuscript.  

Reviewer 3 Report

The manuscript "Pharmacological effects of guava seed polysaccharides: 3 GSF-3 inhibits PC-3 prostate cancer cell growth through 4 immunotherapy in vitro" has all the requirements to be published in Int. J. Mol.

In the first place, the authors have followed the journal's publication rules in detail.

The summary is clear and does the job.

The introduction is also offers us a situation of the state of the art that is treated in this work.

The material and method is well written and very complete. I wish to emphasize at this point that the authors have collected material for a herbarium. This process is essential for any work on phytochemistry and biological activity since in this way we can ensure a correct identification of the species under study. And also, in this way we can allow the study to be replicated to other researchers. All this process is lacking in some phytochemical works and in some cases they are published. If it had been in my hands, I would not have given them the go-ahead.

Regarding the results, they are very well presented, highlighting that they have been presented in several sections for better understanding.

The discussion seems correct to me but in this case it would have facilitated its reading, and it would have been more enriching if it had been subdivided into the same sections as the results.

Some specific aspects to correct:

- Some species are cited with the author, such as Psidium guajava L. Then all the species cited in the text must have the author throughout the text (Ganoderma lucidum, Lycium barbarum, Camellia sinensis, etc ... ..)

- Line 101: Psidium guajava L.inn

- Keyword: add guava seed, Psidium guajava, prostate cáncer

- Include in the title Psidium guajava L.

Author Response

Response to Reviewer 2 Comments

Ms. Ref. No.: ijms-1142027

Point 1: The manuscript "Pharmacological effects of guava seed polysaccharides: GSF-3 inhibits PC-3 prostate cancer cell growth through immunotherapy in vitro" has all the requirements to be published in Int. J. Mol.

In the first place, the authors have followed the journal's publication rules in detail.

The summary is clear and does the job.

The introduction is also offers us a situation of the state of the art that is treated in this work.

The material and method is well written and very complete. I wish to emphasize at this point that the authors have collected material for a herbarium. This process is essential for any work on phytochemistry and biological activity since in this way we can ensure a correct identification of the species under study. And also, in this way we can allow the study to be replicated to other researchers. All this process is lacking in some phytochemical works and in some cases they are published. If it had been in my hands, I would not have given them the go-ahead.

Regarding the results, they are very well presented, highlighting that they have been presented in several sections for better understanding.

The discussion seems correct to me but in this case it would have facilitated its reading, and it would have been more enriching if it had been subdivided into the same sections as the results.

Response 1: Pages 27 - 28 (revised manuscript): We thank the Reviewer’s positive comments and kindly suggestions. The discussion section has been rearranged as the section of results. The first paragraph in the discussion section has been moved to the third paragraph in the revised manuscript. Responses to the Reviewer’s comments together with the itemized changes have been made.

Point 2: Some specific aspects to correct:

- Some species are cited with the author, such as Psidium guajava L. Then all the species cited in the text must have the author throughout the text (Ganoderma lucidum, Lycium barbarum, Camellia sinensis, etc ... ..)

Response 2: The species name cited in the text has been added the author throughout the text.  

Point 3: - Line 101: Psidium guajava L.inn

Response 3: The words “Psidium guajava Linn” have been corrected to “Psidium guajava L.” in the revised manuscript.

Point 4: - Keyword: add guava seed, Psidium guajava, prostate cancer

Response 4: We thank the Reviewer’s suggestion. The keywords have been added into the revised manuscript.

Point 5: - Include in the title Psidium guajava L.

Response 5: We thank the Reviewer’s suggestion. The title has been added “Psidium guajava L.” in the revised manuscript. The title has been modified to “Pharmacological effects of guava (Psidium guajava L.) seed polysaccharides: GSF3 inhibits PC-3 prostate cancer cell growth through immunotherapy in vitro”.

Round 2

Reviewer 2 Report

The comment previously posted is not addressed